# Electron-Transferring Flavoprotein and Its Dehydrogenase Required for Fungal Pathogenicity in *Arthrobotrys oligospora*

**DOI:** 10.3390/ijms252010934

**Published:** 2024-10-11

**Authors:** Yonglan Liu, Zhangyu Li, Junjie Liu, Xiqi Zhang, Xin Wang

**Affiliations:** State Key Laboratory for Conservation and Utilization of Bio-Resources in Yunnan, Yunnan University, Kunming 650091, China

**Keywords:** electron transfer flavoprotein, nematode–fungus interaction, trap

## Abstract

Electron transfer flavoprotein (ETF) plays an important function in fatty acid beta oxidation and the amino acid metabolic pathway. It can provide pathogenicity to some opportunistic fungi via modulating cellular metabolite composition. *Arthrobotrys oligospora* is a typical invasion fungus to nematodes. Its ETF characterization is still unknown. Here, we showed that the mutations of *A. oligospora* ETF (*Aoetfα* and *Aoetfβ*) and its dehydrogenase (*Aoetfdh*) led to severe defects in mitochondrial integrity and blocked fatty acid metabolism. The pathogenicity-associated trap structures were completely suppressed when exposed to nematode-derived ascarosides and nutrition signals, including ammonia and urea. Compared to the wild-type strain, the nematode predatory activity was significantly reduced and delayed. But surprisingly, the rich nutrition could restore the massive trap and robust predatory activity in the mutant *Aoetfβ* beyond all induction cues. Moreover, the deletion of *Aoetfβ* has led to the accumulation of butyrate-like smell, which has a strong attraction to *Caenorhabditis elegans* nematodes. Ultimately, ETF and its dehydrogenase play a crucial role in nematode-trapping fungi, highlighting mitochondrial metabolite fluctuations that are connected to pathogenesis and further regulating the interactions between fungi and nematodes.

## 1. Introduction

Electron transfer flavoproteins (ETF) transfer electrons from mitochondrial dehydrogenases, such as acyl-CoA dehydrogenases, to the electron transport chain, which facilitates the production of ATP. This protein is conserved across a wide range of organisms and exists as a heterodimeric complex, consisting of ETFα and ETFβ subunits. ETF complexes are vital to fatty acid β-oxidation and amino acid metabolism. Both bacteria and eukaryotes contain ETF, that binds FAD and AMP as cofactors [1]. Flavoproteins rely on ETF to transfer electrons to the respiratory chain. Extensive research has been conducted on ETF in yeast, animals, and humans [2,3,4]. Mutations in the *ETFα* or *ETFβ* genes can lead to multiple acyl-CoA dehydrogenase deficiency (MADD) and other metabolic disorders [5]. Similarly, in plants, these mutations result in the accumulation of reactive oxygen species (ROS), which subsequently damage cellular components and disrupt plant growth and development [6]. It is therefore crucial to understand the function of ETF in order to gain a deeper understanding of cellular energy metabolism and diseases related to it.

Nematode-trapping fungi (NT fungi) are critical biological factors in regulating nematode populations within soil ecosystems [7,8,9]. These fungi capture, eliminate, and digest nematodes by forming various predatory structures, including adhesive hyphal networks, adhesive knobs, and constricting rings [10,11,12,13]. Signals, toxins, and effector proteins produced by fungi directly impact nematodes at each process of the interaction [14,15]. Approximately 380 species of nematode-trapping fungi have been identified, with most belonging to the genera *Arthrobotrys*, *Cystopage*, *Dactylellina*, *Dactylella*, *Drechslerella*, *Hohenbuehelia*, and so on [16].

After nematodes are captured by *A. oligospora*, they provide nutrients that facilitate the growth of more hyphae and spores [17]. Early studies indicate that nematode-derived nutrients are first converted into lipids by the fungus and then degraded via the β-oxidation pathway to support new hyphal growth [18]. During this process, fatty acid length determines whether degradation occurs in the mitochondria or peroxisomes: fatty acids with fewer than 20 carbons are oxidized in the mitochondria, while those with more than 20 carbons are initially oxidized in peroxisomes into shorter fatty acids, which are then fully oxidized in mitochondria [19]. The first dehydrogenation step in mitochondrial β-oxidation is catalyzed by acyl-CoA dehydrogenase, which transfers electrons to ETF. Then ETF transfers electrons to ubiquinone oxidoreductase through ETF dehydrogenase (ETFdh) in the respiratory chain, where electrons are transferred to coenzyme Q10 in the mitochondrial inner membrane for ATP synthesis [20,21,22]. Besides electron transport, ETF participates in aerobic metabolism and the production of superoxide and hydrogen peroxide. Mutations in *ETF* are linked to fatty acid oxidation disorders, such as multiple acyl-CoA dehydrogenase deficiency [23,24]. In summary, ETF plays a critical role in electron transfer and energy synthesis.

The research on ETF in filamentous fungi is limited. In this study, we identified the functions of *etf* and *etfdh* in *A. oligospora* through gene knockout. Our results demonstrated that both *etf* and *etfdh* are involved in the formation of traps in *A. oligospora*, particularly highlighting the strong attraction of *Caenorhabditis elegans* to the Δ*Aoetfβ* mutant. In summary, our findings suggest that ETF and ETFdh contribute to predator–prey interactions between predatory fungi and nematodes, providing a theoretical basis for understanding these interactions.

## 2. Results

### 2.1. Analysis and Deletion of etf and etfdh in A. oligospora

Based on the genome annotation data of *A. oligospora* ATCC24927, we obtained probable electron transfer flavoprotein coding genes, *Aoetfα* (AOL_s00109g97), *Aoetfβ* (AOL_s00078g521), and ETF dehydrogenase coding gene *Aoetfdh* (AOL_s00215g393). A conserved domains analysis showed us that both subunits of ETF-α and ETF-β contain typical ETF domains (Figure 1A). Further modeling revealed that ETF-α associates with ETF-β to form a stable heterodimeric complex, designated AoETF. This complex binds both the FAD cofactor and an AMP molecule (Figure 1B). In contrast, ETFdh functions as a monomer and contains several cofactor binding sites for NAD, ETFQO_UQ, and an iron-sulfur cluster (4Fe-4S) (Figure 1A). So far in NT fungi, the roles of ETF heterodimer and ETFdh have not been characterized. Phylogenetic tree analysis revealed that ETF as well as ETFdh are highly conserved in NT fungi with other fungal species, suggesting that they may have similar functions in this group of fungi (Appendix A). Using the homologous recombination strategy, we generated mutants for each of these three coding genes, designated as Δ*Aoetfα*, Δ*Aoetfβ*, and Δ*Aoetfdh* (Figure 1C,D). To verify protoplast transformants using PCR, the sizes of the wild-type and mutant strains for gene *Aoetfα* are 1425 bp and 2563 bp, respectively. For gene *Aoetfβ*, the sizes of the wild-type and mutant strains are 1142 bp and 2353 bp, respectively. For gene *Aoetfdh*, the sizes of the wild-type and mutant strains are 1564 bp and 2460 bp, respectively (Appendix A). On various nutrient-rich media, including PDA, CMY, TG, and TYGA, the vegetative growth of the three mutants was comparable to that of the wild-type strain (Appendix A).

### 2.2. Aoetf and Aoetfdh Impair Conidiation in A. oligospora

We initially assessed whether *Aoetf* and *Aoetfdh* are involved in the sporulation of *A. oligospora*. Wild-type and mutant strains were cultivated on PDA medium for six days, after which conidial density and spore number were measured. The results showed a significant decrease in both conidial density and spore number in the Δ*Aoetfα*, Δ*Aoetfβ*, and Δ*Aoetfdh* compared to the wild-type (Figure 2A,B). Regarding conidial morphology, both wild-type and mutant strains produced four main types of conidia, consistent with previous findings [25]. Among these, the second type, representing mature spores in the phialides, was predominant in both wild-type and mutant strains (64%, 59%, 63.67%, and 62.33%, respectively) (Figure 2C,D). However, the deletion of Δ*Aoetfα* resulted in a notably low proportion of immature spores, constituting only 1.33% of the total. These observations suggest that the *Aoetf* and *Aoetfdh* have important involvement in conidiation in *A. oligospora.* Unexpectedly, the Δ*Aoetfβ* strain was found to spontaneously produce nematode-capturing structures, adhesive three-dimensional networks, when grown on CMY medium (Figure 2E), which contrasts with the typical formation of predatory structures only under low-nutrient conditions.

### 2.3. Aoetf and Aoetfdh Are Essential for the Formation of Traps in A. oligospora

To investigate whether *Aoetf* and *Aoetfdh* influence signal-induced trap formation in *A. olilgospora*, we compared trap production between wild-type and mutant strains in response to live *C. elegans*, nematode extracts, and trap-inducing signals such as ascaroside (ascr#7), urea, and ammonia (Am). Our findings revealed that the disruption of *Aoetf* or *Aoetfdh* genes delayed trap formation and reduced nematode-killing efficiency compared to the wild-type strain under same conditions. Specifically, when exposed to nematodes, the mutant strains produce traps later than the wild-type strains (Figure 3A). In the wild-type strain, only 41.67% of nematodes survived after 12 h, while the survival rates in the Δ*Aoetfα*, Δ*Aoetfβ*, and Δ*Aoetfdh* strains were 71.43%, 85.71%, and 63.49%, respectively (Figure 3B). Furthermore, when exposed to whole nematode extracts, the mutants produced fewer traps than the wild-type strain (Figure 3C). Notably, when we tested trap induction in response to nematode-derived ascaroside, urea, and ammonia, the mutants completely failed to produce any traps (Figure 3D). These results suggest that mutations in *Aoetf* or *Aoetfdh* significantly impair trap formation in *A. oligospora*.

### 2.4. The Absence of Aoetf Affects Fatty Acid Metabolism in A. oligospora

In other organisms, ETF plays crucial roles in fatty acid metabolism, amino acid metabolism, and mitochondrial function. To determine whether *Aoetf* in *A. oligospora* has similar effects, we analyzed hyphal growth in wild-type and mutant strains on MM medium supplemented with sodium carbonate and four different fatty acids (sodium butyrate, caprylic acid, lauric acid, and oleic acid) as the sole carbon source. The results indicated that the growth of the mutant strains was significantly inhibited, with lauric acid completely blocking their growth. These findings suggest that *Aoetf* and *Aoetfdh* are essential for fatty acid metabolism in *A. oligospora* (Figure 4A). Furthermore, transcriptomic sequencing revealed that the *Aoetf* mutation led to significant changes of nearly 60 genes related to lipid metabolism in expression level; the metabolism pathways involving glycerophospholipid metabolism, steroid biosynthesis, fatty acid degradation, and biosynthesis of unsaturated fatty acids were significantly enriched (Figure 4B–D). Therefore, mutations in ETFs have a significant impact on intracellular fatty acid metabolism in fungi, which can inhibit the degradation of long-chain fats such as lauric acid by fungi.

### 2.5. Aoetf Deletion Impairs Mitochondrial Function in A. oligospra

We guessed the defects in sporulation and pathogenicity are attributed to impaired mitochondrial function, so we conducted transmission electron microscopy (TEM) observation. The TEM results revealed significant mitochondrial damage in the mutant strains compared to the wild-type strain. Specifically, the mitochondria in the Δ*Aoetfα* strain exhibited a complete loss of cristae, while those in the Δ*Aoetfβ* strain displayed fragmented cristae (Figure 5A–C). Additionally, mitochondrial staining demonstrated that the wild-type strain had well-defined, prominent mitochondria, whereas the Δ*Aoetfα* and Δ*Aoetfβ* strains showed reduced staining intensity (Figure 5D–F), indicating a decrease in mitochondrial viability due to the mutations.

### 2.6. The Aoetf Mutant Strain Has the Ability to Attract C. elegans

ETF dysfunction has been implicated in a variety of metabolic disorders involving fatty acids and amino acids, particularly in the context of glutaric acidemia in humans [26]. During the growth of the mutant strains, a strong odor reminiscent of sweaty feet was detected. Considering the well-established attraction of NT fungi to nematodes, we conducted chemotaxis assays using the Δ*Aoetfβ* strain and the wild-type strain in the presence of *C. elegans*. The results showed that Δ*Aoetfβ* has a significant attraction to *C. elegans*, with a chemotaxis index of 0.92 (Figure 6A). Further, we evaluated the chemotactic response of *C. elegans* to glutaric acid and observed that this compound exhibited a higher attraction index of 0.8, significantly greater than that of the control (Figure 6B). In the metabolomics analysis, several significantly upregulated compounds were observed (Table 1, Appendix A), among which the markedly upregulated compound cetraxate demonstrated an attraction to *C. elegans* with a value of 0.70 (Figure 6C). Given the role of fungal extracellular vesicles (exosomes) as mediators for shuttling chemical signals to communicate with host cells [27], we isolated exosomes from both the wild-type and Δ*Aoetfβ* strains for further attraction assays. Nanoparticle Tracking Analysis (NTA) detection and TEM imaging revealed that the exosomes from the mutant strains were larger in size and present in higher concentrations compared to those from the wild-type strain (Appendix A). Chemotaxis assays with exosomes demonstrated that exosomes from both the wild-type and Δ*Aoetfβ* strains could attract *C. elegans*; however, the exosomes from the mutant strain exhibited a higher attraction index of 0.81, compared to 0.63 for the wild-type exosomes (Figure 6D,E). Furthermore, a comparison of the relative chemotactic strength revealed that the Δ*Aoetfβ* strain had a significantly higher chemotaxis index of 0.73 compared to 0.26 for the wild-type strain (Figure 6F). Based on these observations, we conclude that the *Aoetf* mutation leads to the accumulation of metabolites with nematode-attracting activity, which may be secreted via fungal extracellular vesicles.

### 2.7. Nematode ETF Exhibited Similar Roles in Mitochondrial Function to A. oligospora

Nematode-NT fungi are consistently present in soil; we wondered about the impact of homologous genes of *A. oligospora* ETF on the mitochondrial function of nematodes. Based on the observation of the TEM, *C. elegans* exhibited mitochondrial swelling and cristae disruption when the ETF coding was interfered with (Figure 7A). Additionally, the behavior assay showed that obvious aggregation, nearly 100%, occurred following the interference of ETFα, while the aggregation ratio of ETFβ was 60% compared to N2 *C. elegans* (Figure 7B).

## 3. Discussion

ETF and ETFdh are highly conserved across bacteria, filamentous fungi, and animals. In our study, AoETF and AoETFdh also exhibited a high degree of conservation with other filamentous fungi (Figure 1 and Appendix A). We established multiple roles of *Aoetf* and *Aoetfdh* in the phenotypic regulation of conidiation and pathogenesis in *A. oligospora* (Figure 2 and Figure 3). In the *Aoetf* mutant strains, transcriptional profiles and metabolic omics revealed distinctive differences in fatty acid metabolism and amino acid metabolism, which were further confirmed by the growth defect against long chain fatty acids (Figure 4, Table 1). Similar results have been previously demonstrated in the rice blast fungus *Magnaporthe oryzea* and in the entomopathogenic fungus *Beauveria bassiana* [2,3].

In humans, *ETF* and *ETFdh* variants are typical causes of multiple acyl-CoA dehydrogenase deficiency (MADD) and glutaric aciduria type II [28]. In cases of ETF or ETFdh deficiency, the acyl-CoA dehydrogenases cannot transfer electrons from several dehydrogenases involved in fatty acid oxidation, as well as choline and amino acid metabolism [29]. This process would lead to the deficiencies of multiple acyl-CoA dehydrogenases and disruption of ATP production, causing a large accumulation of glutaric acid, lactic, ethylmalonic, butyric, isobutyric, 2-methyl-butyric, and isovaleric acids [30,31]. As we know, these compounds are volatile and emit unpleasant odors. Coincidentally, our *A. oligospora* mutant of *Aoetf* also produced a similar foul smell, which we suspected was glutaric acid. For fungi to capture nematodes, fatal attraction is an important process. In this study, we found that glutaric acid was strongly attracted to *C. elegans*, although we did not obtain exact compound information based on metabolomics (Figure 6). Moreover, the mutant and its exosomes showed stronger attraction to nematodes than the wild-type strain, suggesting that the disruption of *Aoetf* led to the secretion of certain functional metabolites (Figure 6). Therefore, the metabolic regulatory role of *Aoetf* and *Aoetfdh* in the fungus *A. oligospora* may be evolutionarily conserved with that in higher animals.

Few studies have described the integrity of mitochondrial morphology in ETF-ETFdh deletions [3,32]. In this study, we found that metabolic disorder-produced esters and acids might impair mitochondrial fitness in *A*. *oligospora* (Table 1). TEM images showed defective mitochondria with fragmented, cristae-free, or swollen mitochondria exist in the mutants of *Aoetf* (Figure 5). Since mitochondria are the key organelles for energy production, mutations or deletions in ETF and ETFdh in *Magnaporthe oryzae* lead to reduced ATP levels, affecting its energy metabolism, which may further impact its growth, development, and pathogenicity [3]. Based on these observations, mitochondrial dysfunction may be the reason why fungi fail to respond to signals to produce predatory organs. In most eukaryotic cells, mitochondria are responsible for the delicate metabolic activities and stress responses that maintain cellular stability [33,34]. Our research provides the possibility that inhibition of mitochondrial function could be a potential strategy for combating fungal infections in nematodes.

We have identified the multifaceted functions of *Aoetf* and *Aoetfdh* that include maintaining mitochondrial structure, fatty acid metabolism, and further controlling pathogenicity and sporulation. The lifestyle of NT fungi includes vegetative growth and parasitic growth, but the latter only occurs in nutrient-deprived conditions [35]. Unexpectedly, we identified a mutant strain of *Aoetfβ* that grew autonomously in the form of trapping hypha under rich nutrition, which can actively kill nematodes (Figure 2). In addition, *Aoetf* mutations acquired a strong ability to attract nematodes. From the perspective of the host nematode, *Aoetf* also plays a conservative role in maintaining mitochondrial integrity and has a significant regulatory effect on nematode aggregation behavior (Figure 7). This *Aoetf* mutation-induced change in animal behavior has been observed in other animal models, where ETF-resulted metabolic dysfunction is associated with aberrant neural phenotypes and paralysis [36]. The significance of these findings should further facilitate mutant strains’ potential in controlling nematodes in the field.

Faced with low or poor nutrition, the NT fungi can respond to context signals to form infection structures to capture and digest nematodes [35,37]. The involved signaling substances currently cover a wide range, including nematode or nematode-derived chemicals and nutrition signals like urea or ammonia [10,38,39]. The G protein signaling pathway plays an especially vital role for trap formation in NT fungi [39,40]. Unexpectedly, we discovered that changes in intracellular metabolic or energy levels induced by genetic manipulation might activate the regulation of trap formation in *A. oligospora*. Our future work will aim to link cellular metabolic fluctuations to signal transduction, extending the understanding of intracellular signal regulation to trap development and pathogenesis in the NT fungi.

## 4. Materials and Methods

### 4.1. Strains, Plasmids, and Culture Conditions

The wild-type (WT) strain of *A. oligospora* (ATCC 24927) and the mutant strains *Aoetfα*, *Aoetfβ*, and *Aoetfdh* were inoculated on PDA medium (200 g potato, 20 g glucose, and 18 g agar) and incubated at 28 °C. The plasmids pCSN44 and pCE-Zero (Vazyme Biotech Co., Ltd., Nanjing, China) were used for the amplification of the hygromycin fragment and the construction of knockout vectors, respectively. *Escherichia coli* strain (DH5α) (TransGen Biotech, Beijing, China) was used for plasmid cloning and storage, grown in LB medium at 37 °C. ssPDA medium (PDA supplemented with sucrose) was used for protoplast regeneration, while CMY medium (cornmeal-molasses-yeast), TG medium (tryptone-glucose), and TYGA medium (tryptone-yeast-extract-glucose-agar) were used to analyze the phenotypic characteristics of hyphae and spores [41]. *C. elegans* was cultured on NGM medium at 20 °C for bioassays.

### 4.2. Targeted Gene Deletion

The sequences of *Aoetfα* (AOL_s00109g97), *Aoetfβ* (AOL_s00078g521), and *Aoetfdh* (AOL_s00215g393) were obtained from the *A. oligospora* genome in the NCBI database. Primers for the upstream fragment, hygromycin fragment, and downstream fragment were designed using the NEB England online tool (https://nebuilderv1.neb.com). Paired primers were used to amplify the upstream and downstream fragments of *Aoetfα*, *Aoetfβ*, and *Aoetfdh* from *A. oligospora* by PCR, and the hygromycin fragment was amplified using the pCSN44 plasmid as a template. The pCE-Zero vector was digested with *Eco*RV, and the three fragments were ligated to the vector and transformed into *E. coli* competent cells. Primers for gene knockout are listed in Appendix A. Subsequently, the fragments were transformed into *A. oligospora* protoplasts using the PEG/CaCl_2_-mediated method. Positive transformants were selected on TYGA medium containing 100 μg/mL hygromycin and verified by PCR.

### 4.3. Phenotypic Characterization of Strains

The wild-type and mutant strains were inoculated on PDA, CMY, TYGA, and TG media, and the colony diameters were measured every 24 h using a vernier caliper for a duration of 6 days. To determine conidial production, the wild-type and mutant strains were inoculated on CMY medium and cultured for 7 days. The spores were washed with 5 mL of double-distilled water, and 10 µL of the spore suspension was taken for counting using an optical microscope, with each experiment repeated three times [42]. Furthermore, different fatty acids were used as the sole carbon source to observe differences in fatty acid metabolism in the mutant strains. To observe conidia, the wild-type and mutant strains were inoculated on PDA medium and cultured for three days, then transferred to water agar plates for 24 h and observed under an optical microscope (Olympus, Tokyo, Japan).

### 4.4. Microscopic Observation

To observe the mitochondria, the hyphal cells were stained with 10 μg/mL mitochondrial staining (Mito Tracker™ Red. Invitrogen, Thermo Fisher Scientific, Waltham, MA, USA), and images were captured using a confocal laser scanning microscope (Zeiss, Oberkochen, Germany). Traps produced on nutrient-rich medium were scanned using a cryo-scanning electron microscope. Mycelia cultured on CMY were collected and fixed with glutaraldehyde electron microscopy fixation solution, then sent to the Kunming Institute of Zoology, Chinese Academy of Sciences, for transmission electron microscopy imaging. The interfered nematodes were fixed with glutaraldehyde and imaged using Transmission Electron Microscope (TEM) (Zeiss, Oberkochen, Germany).

### 4.5. Difference in Trap Formation

Spore suspensions containing 2000–3000 spores of the wild-type and mutant strains, cultured for 7 days, were spread onto WA plates and grown for 30 h. Different compounds (ascr#7, urea, and ammonia) were added to induce trap formation. The number of traps was counted, and photographs were taken 48 h later. Similarly, 1000 *C. elegans* nematodes were added to induce trap formation in the wild-type and mutant strains, and the traps were photographed and counted at 6, 12, and 24 h.

### 4.6. Chemotaxis Assay

The mutant and WT strains were inoculated on CMY solid and liquid media and cultured for 7 days. *C. elegans* nematodes were synchronized and allowed to mature to adulthood beforehand. Blocks of mutant and wild-type cultures, along with their fermentation broth, were placed on opposite sides of 6 cm NGM agar plates (without peptone), with 10 µL drops of nematode suspension (approximately 200 nematodes) added in the center. The direction of nematode movement was observed, and the number of nematodes moving toward the experimental group and the control group was counted, respectively. Perform a behavioral assay on the interfered nematodes, as described previously. Primers for gene interference are listed in Appendix A [43].

### 4.7. Exosome Assay

The mutant and WT strains were inoculated into TG medium and cultured for 7 days. Subsequently, the culture supernatant was collected and filtered sequentially through 0.45 µm and 0.22 µm filters. The filtered supernatant was subjected to centrifugation at 4 °C for 30 min at 100,000× *g*; the supernatant was discarded, and the pellet was collected. After resuspension in PBS, exosomes derived from mutant and wild-type strains were separately applied to opposite sides of 6 cm NGM agar plates (without peptone), with 10 µL drops containing approximately 200 adult-stage nematodes each added to the center. Nematode movement and direction were observed and quantified. Furthermore, the isolated exosomes were sent to Yanzai Biotechnology (Shanghai, China) Co., Ltd. for imaging and particle size measurement.

### 4.8. Transcriptome and Metabolome

The mutant and wild-type strains were cultured in CMY liquid medium for 7 days. Mycelia and fermentation broth were collected separately, flash-frozen in liquid nitrogen, and sent to Shanghai Majorbio Bio-Pharm Technology Co., Ltd. (Majorbio, Shanghai, China). Gene Ontology (GO) function and Kyoto Encyclopedia of Genes and Genomes (KEGG) pathway enrichment were analyzed using the Majorbio cloud platform (https://www.majorbio.com. accessed on 19 February 2025).

## 5. Conclusions

This study highlights the complications of ETF and its dehydrogenase ETFdh in the pathogenesis and sporulation of NT fungus *A. oligospora*. Further, the disruptions of *Aoetf* and *Aoetfdh* can alter metabolic and energy levels, possibly leading to mitochondrial dysfunction and modifying the interaction between NT fungi and nematodes. Moving forward, future research should focus on the regulatory mechanisms by which metabolic alterations influence signal transduction and phenotypic regulation in the NTF group.

## Figures and Tables

**Figure 1 ijms-25-10934-f001:**
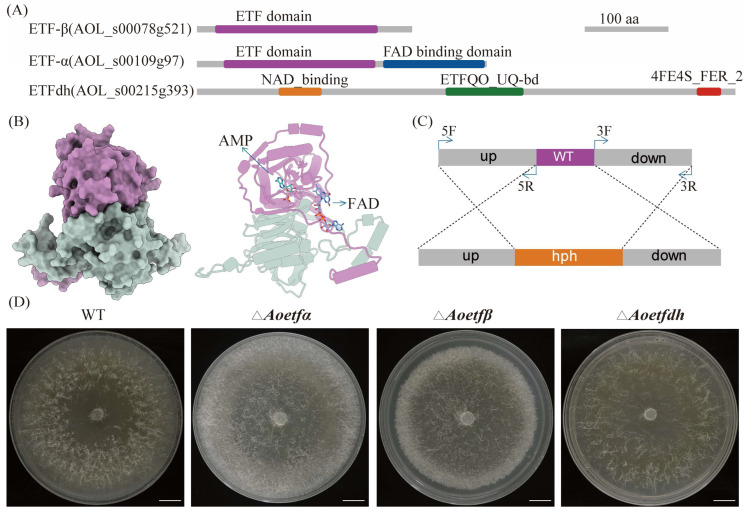
Bioinformatics analysis of ETF, ETFdh and gene knockouts. (**A**) The domain structure of *Arthrobotrys oligospora* ETF-α, ETF-β and ETFdh as annotated at the Broad Institute of MIT. (**B**) Structural Model of ETFα-ETFβ complex (AoETF) bound to AMP and FAD, the purple structure represents ETF-β, while the light gray structure depicts ETF-α. (**C**) Homologous recombination knockout model diagram. (**D**) Growth of wild-type strain (WT) and the mutants of *Aoetfα*, *Aoetfβ*, and *Aoetfdh* on PDA medium for 7 days; scale bar: 1 cm.

**Figure 2 ijms-25-10934-f002:**
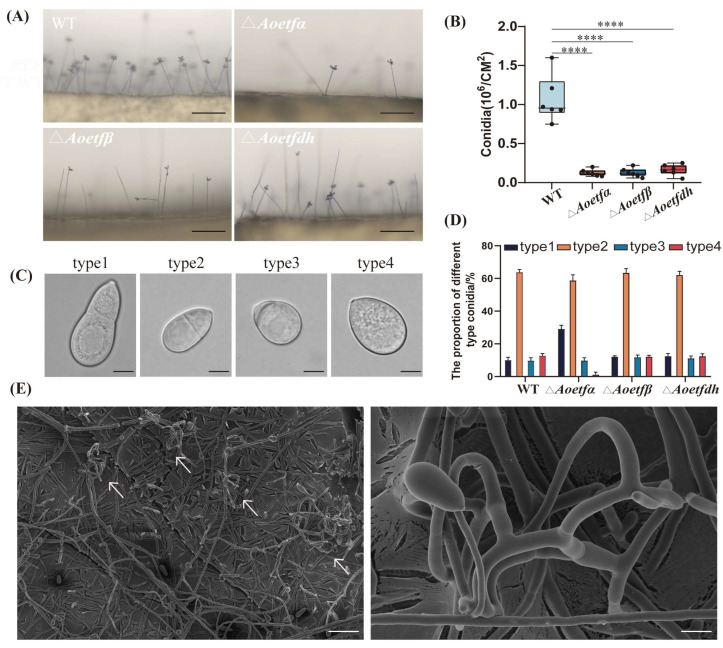
Mutations *Aoetf* and *Aoetfdh* affect spore production, morphology, and trap formation. (**A**) Comparison of conidia in wild-type, Δ*Aoetf*, and Δ*Aoetfdh* strains (Scale bar: 100 μm). (**B**) Comparison of conidia production in wild-type, Δ*Aoetf*, and Δ*Aoetfdh* strains. **** represents significant difference (*p* < 0.0001). (**C**,**D**) Analysis and statistics of conidia types in wild-type, Δ*Aoetf*, and Δ*Aoetfdh* strains (scale bar: 7 μm). (**E**) Scanning electron micrographs of the trapping structure in the Δ*Aoetfβ* strain at different magnifications. The left image is at 150× magnification (scale bar: 100 μm; white arrow indicates the trap), and the right image is at 1000× magnification (scale bar: 10 μm).

**Figure 3 ijms-25-10934-f003:**
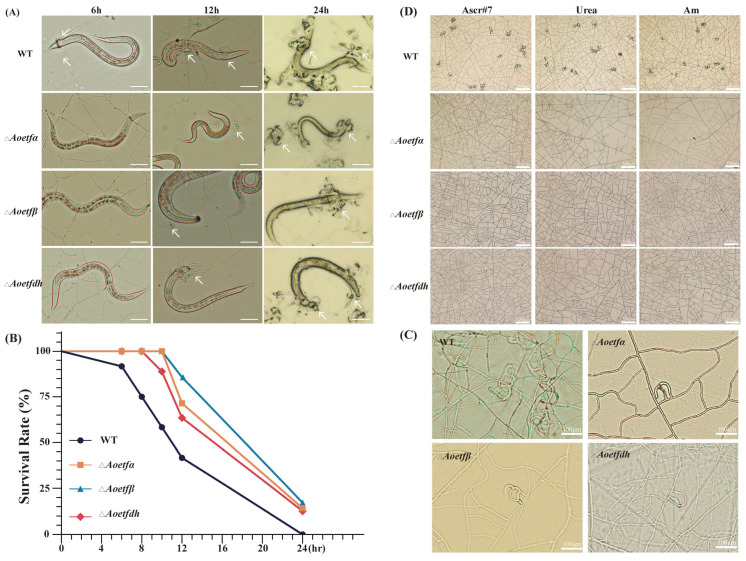
Trap formation in wild-type, Δ*Aoetf*, and Δ*Aoetfdh* strains. (**A**) Representative bright-field images of trap formation at different time points after nematode induction (scale bar: 200 μm; the arrows denote trap structures). (**B**) Survival rate of nematodes exposed to wild-type, Δ*Aoetfα*, Δ*Aoetfβ*, and Δ*Aoetfdh* strains. (**C**). Representative bright-field images of traps induced by nematode extracts in wild-type, Δ*Aoetfα*, Δ*Aoetfβ*, and Δ*Aoetfdh* strains (scale bar: 200 μm). (**D**) Representative bright-field images of traps induced by ascr#7, urea, and ammonia (Am) in wild-type, Δ*Aoetfα*, Δ*Aoetfβ*, and Δ*Aoetfdh* strains (scale bar: 100 μm).

**Figure 4 ijms-25-10934-f004:**
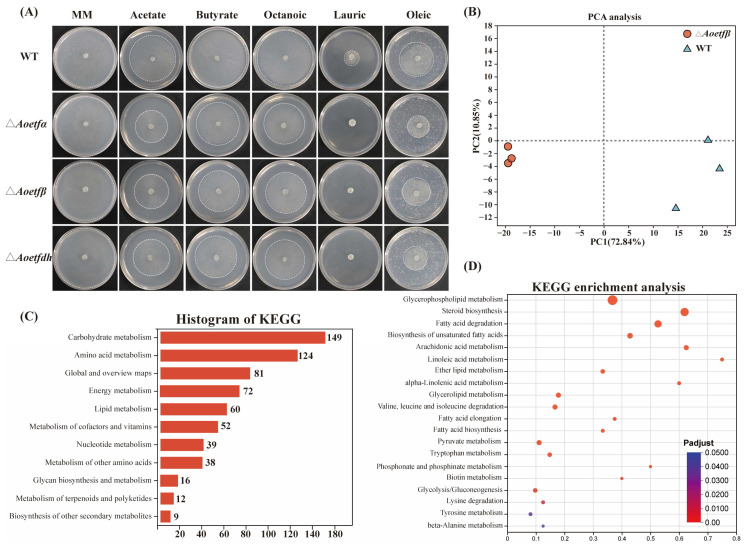
Δ*Aoetf* and Δ*Aoetfdh* were involved in fatty acids metabolism. (**A**) Growth conditions of WT, Δ*Aoetfα*, Δ*Aoetfβ*, and Δ*Aoetfdh* strains on MM medium with various fatty acids as the sole carbon source. (**B**) Principal component analysis (PCA) plot of *Aoetfβ* mutant and wild-type strains. (**C**) Histogram of KEGG-enriched metabolic pathways in the transcriptome, with 60 genes associated with lipid metabolism. (**D**) KEGG enrichment analysis of genes related to lipid metabolism.

**Figure 5 ijms-25-10934-f005:**
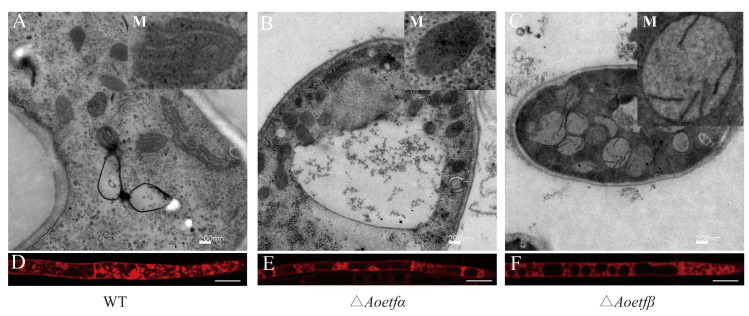
*Aoetf* deletion impairs mitochondria function. (**A**–**C**) Ultrastructural analysis of wild-type, Δ*Aoetfα*, and Δ*Aoetfβ* strains using transmission electron microscopy (scale bars: 200 nm; M denotes mitochondria). (**D**–**F**) Comparison of mitochondrial of wild-type and Δ*Aoetf* strains were stained with MitoTracker^®^ Red (scale bars: 3 μm).

**Figure 6 ijms-25-10934-f006:**
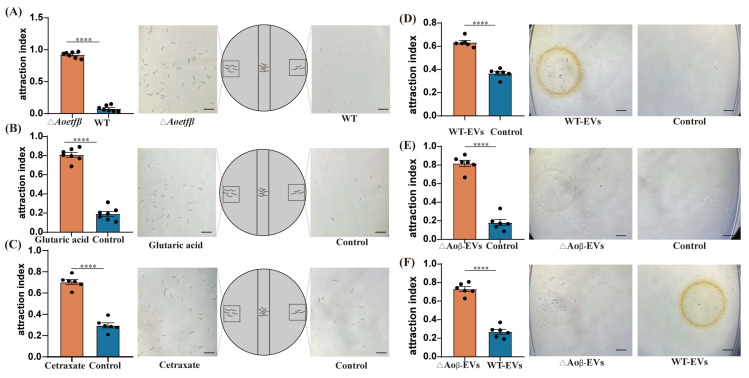
Chemotaxis observation of *C. elegans*. (**A**) Chemotaxis statistics of *C. elegans* towards fermentation broth of wild-type and Δ*Aoetfβ* strains. (**B**) Chemotaxis statistics of *C. elegans* towards glutaric acid and control. (**C**) Chemotaxis statistics of *C. elegans* towards cetraxate and control (**D**) Chemotaxis statistics of *C. elegans* towards exosomes from wild-type strains. (**E**) Chemotaxis statistics of *C. elegans* towards exosomes from Δ*Aoetfβ* strains. (**F**) Comparative chemotaxis statistics of *C. elegans* towards exosomes from wild-type and Δ*Aoetfβ* strains. **** represents significant difference (*p* < 0.0001; scale bars: 2 mm).

**Figure 7 ijms-25-10934-f007:**
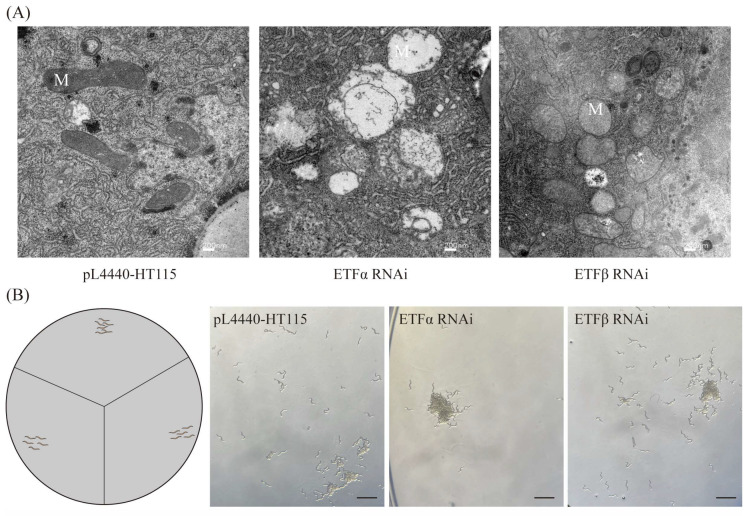
TEM observations and phenotypes of *ETF* Genes in *C. elegans*. (**A**) Transmission electron microscopy images of mitochondria after interfering with the *ETF* gene in *C. elegans* (scale bars: 200 nm). M denotes mitochondria. (**B**) Aggregation phenomenon observed in *C. elegans* after interfering with the *ETFα* gene (scale bars: 2 mm).

**Table 1 ijms-25-10934-t001:** Metabolomics analysis of differentially upregulated compounds between wild-type and Δ*Aoetfβ* strains.

Metabolism Name	FC	KEGG Pathway Description
Cetraxate	47.8841	-
S-Lactoylglutathione	1.4921	Pyruvate metabolism
Kynurenic acid	1.2806	Tryptophan metabolism
Gamma-glutamyl-L-putrescine	1.3035	Arginine and proline metabolism
2-Isopropylmalic acid	1.1205	Biosynthesis of secondary metabolites
		Valine, leucine and isoleucine biosynthesis
		2-Oxocarboxylic acid
Theophylline	1.1008	Biosynthesis of secondary metabolites
		Caffeine metabolism
6-Lactoyltetrahydropterin	1.1103	Folate biosynthesis
Homogentisic acid	1.1126	Ubiquinone and other terpenoid-quinone biosynthesis
		Tyrosine metabolism
		Biosynthesis of cofactors
Imidazole lactic acid	1.1166	Histidine metabolism
D-xylonic acid	1.0826	Pentose and glucuronate interconversions
		Ascorbate and aldarate metabolism
N-carbamoylputrescine	1.0977	Arginine and proline metabolism
5-methoxyindoleacetate	1.0983	Tryptophan metabolism
lactose	1.0749	ABC transporters
		Galactose metabolism
Indole-3-acetaldehyde	1.0808	Tryptophan metabolism
Glucosamine	1.0607	Amino sugar and nucleotide sugar metabolism
		Biosynthesis of nucleotide sugars
LL-2,6-diaminopimelic acid	1.0692	Biosynthesis of secondary metabolites
		D-amino acid metabolism
		Biosynthesis of amino acids
		Lysine biosynthesis
Biotin	1.0596	Biotin metabolism;ABC transporters
		Biosynthesis of cofactors
Kynurenine	1.0558	Tryptophan metabolism
		Biosynthesis of cofactors

FC (Δ*Aoetfβ*/WT): fold change in the expression of this metabolite between the two groups. WT: expression level of this metabolite in the control group WT. Δ*Aoetfβ*: expression level of this metabolite in the experimental group Δ*Aoetfβ*. Enriched KEGG metabolic pathways.

## Data Availability

The original contributions presented in the study are included in the article and Appendix A, further inquiries can be directed to the corresponding author, Xin Wang.

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
