# Peer review of "Electron-Transferring Flavoprotein and Its Dehydrogenase Required for Fungal Pathogenicity in Arthrobotrys oligospora"

_ijms, 2024, doi:10.3390/ijms252010934_

Round 1

Reviewer 1 Report

Comments and Suggestions for Authors

See the files

Reviewer 2 Report

Comments and Suggestions for Authors

In this current study, the authors identified the functions of ETF and ETFdh in A. oligospora through gene knockout. They demonstrated that both ETF are involved in the formation of traps in A. oligospora. Specifically, ΔAoetfβ plays a role in attracting C. elegans. Their findings suggest that ETF contributes to predator-prey interactions between predatory fungi and nematodes, providing a theoretical basis for understanding these interactions.

Some suggestions:

1. Point 2.6. Chemotaxis assay, line 129: please add how did you quantified the movement and orientation of the nematodes.

2. Point 2.7. lines 135-36: give please some details concerning the “high-speed centrifugation” used  to isolate exosomes.

3. Tables S1 and S2 are not presented as supplementary material. Moreover, in my opinion they should be included in the article. 4. Pg 7, line 226 – sodium carbonate isn’t a fatty acid. 5. The description of the methods could be more detailed. 6. Most of the aspects presented at the discussions are taken from the literature. The results obtained in the current study are not properly discussed.

7. Add please a conclusion at the end of the article.

8. Please underline the utility of the study.

9. Unfortunately, you forgot to upload the additional materials regarding the article.

Comments on the Quality of English Language

Minor editing of English language is required.
